



# Sub catchment Assessment of snowpack and snowmelt change by analyzing elevation bands and parameter sensitivity in the high Himalayas

Vishal Singh[1], Manish Kumar Goyal[1], Rao Y. Surampalli[2], and Francisco Munoz-Arriola[3,4]

[1]Department of Civil Engineering, Indian Institute of Technology, Guwahati-781039, India

[2]Global Institute for Energy, Environment, and Sustainability, P.O. Box 14354, Lenexa, Kansas 66285, USA

[3]Department of Biological Systems Engineering, University of Nebraska-Lincoln, USA

[4]School of Natural Resources, University of Nebraska-Lincoln, USA

**Keywords:** Hydrological modeling, Himalayas, SWAT and SWATCUP, Snowpack and snowmelt, elevation bands.

*Correspondence to*: Francisco Munoz-Arriola (fmunoz@unl.edu)

**Abstract**: The present work proposes to improve estimates of how much streamflow is generated by snow in the watersheds of the steep Himalayas. Half of the earth's glacial catchments in nonpolar areas are in the Himalayas, and they generate almost a third of the streamflows in India. In River catchments with glacier presence in the region, temporal variability in streamflow generation and the associated distribution of accumulated snow illustrate how changes in snowmelt and precipitation can affect water supplies to a growing population of 1.3 billion people. Estimations of snowpack and snowmelt in watersheds are critical for understanding streamflow generation and sources of catchments. However, estimating precipitation and snow accumulation is constrained by the difficulties complex terrain poses to data collection. The primary objective of this study is to assess the role of elevations in the computation of snowfall (snowpack) and snowmelt in sub-catchments. The study area is the Satluj River Catchment (up to Kasol gauge) with moderate (e.g., 526 m) to very high elevations (e.g., 7429 m) dominated by snow covers and glaciers. The Satluj River Catchment was divided into 14 sub-catchments. Snowpack and snowmelt variations in the sub-catchments in both historical and projected near-term (2011-2130) periods were analyzed using observed and Global Circulation Model (GCM) data sets. Both hydrological scenarios used elevation bands and parameter-sensitivity analyses built in the Soil Water Assessment Tool (SWAT) model. For model calibration/validation and parameter sensitivity analysis, an advanced optimization method—namely, Sequential Uncertainty Fitting (SUFI2) approach was used with multiple hydrological parameters. Among all parameters, the curve number (CN2) was found significantly sensitive for computations. The snowmelt hydrological parameters such as snowmelt factor maximum (SMFMX) and snow coverage (SNO50COV) significantly affected objective functions such as $R^2$ and NSE during the model optimization process. The computed snowpack and snowmelt were found highly variable over the Himalayan sub-catchments as also reported by previous researchers in other regions. The magnitude of





snowpack change consistently decreases across all the sub-catchments of the Satluj River Catchment (varying
between 4% and 42%). The highest percentage of changes in snowpack was observed over high-elevation
subcatchments.

## 1. Introduction

Most of the perennial river channels such as the Ganga, Indus, and Brahmaputra are originated in Himalayan
glaciers. Large snowpacks along the Himalayas are formed by thousands of glaciers in valleys and are the major
sources of fresh water reserves in India (Bolch et al., 2012). Many studies reported that the hydroclimatology of the
Himalayan catchments is changing, and thus snowpack and glaciers are reducing their mass, which leads to more
snowmelt water into the streams (Bhambri and Bolch, 2009; Bolch et al., 2012; Xu et al., 2016; Singh et al., 2016).
According to the Intergovernmental Panel on Climate Change (IPCC, 2013), changes in temperature and
precipitation are expected to affect the hydrology of Himalayan catchments (IPCC, 2013). Some of these changes
can be reflected in the spatial distribution and temporal variability of rainfall, snowfall and glaciers' mass, which at
the same time can drive streamflow generation in large catchments in the Himalayas (Singh et al., 2008). While
glaciers influence streamflows in high altitudes, rainfall is considered a predominant factor in low altitudes. As a
main tributary of the Indus River, the Satluj River has its flow primarily generated by snowmelt during the spring.
Thus, a higher melting will result in an increase in runoff downstream before the monsoon season (Jain et al., 2010)
and increased vulnerability to floods and risk to the sustainability of agriculture in the Punjab region. Other areas of
the world, such as the western United States of America, have experienced increments in altitude of snow
accumulation reduction of the snowpack, and earlier snowmelt onsets (Motte et al., 2005; Mote, 2006). All of these
factors influence water supply and storage and affect the sustainability of human activities downstream. However, in
the Sutlej River Catchment, recent and projected changes in snowmelt and snowpack are inconclusive about how
glacial and perennial streamflow will be affected in a changing climate.
Several studies highlighted an elevation-dependent warming and revealed that changes in temperature lapse rate
(TLR) and precipitation lapse rate (PLR), due to climate change, are responsible for the higher reduction in the
snowpack at high elevations than those present at lower elevations (Singh and Goyal, 2016a; Singh and Goyal,
2016b). The TLR and PLR are functions of elevation (Gardner et al., 2009), and thus the snowpack and snowmelt
rate can be affected by variations in TLR and PLR as analyzed by Singh and Goyal (2016a and 2016b) over eastern
Himalayan catchments. After temperature, alterations in precipitation pattern have been recognized as another major
factor that determines changes in snowpack over the region. Thus, climate change projections indicate an increment
in precipitation variability (Change, I.C., 2013) which will influence PLRs (Singh and Goyal, 2016a) and snowfall
patterns, particularly when catchments' topography corresponds with moderate to very high elevations like the
Himalayas. Also, influences of a changing climate in the Himalayan regions have evidenced long-term shifts in
average air temperature, precipitation and other land surface variables (Sridhar and Nayak, 2010; Jain et al., 2010;




Beniston, 2012; Narsimlu et al., 2013;). Bolch et al. (2012) reported that the length of many Himalayan glaciers is shortening, and only 25% of glaciers are stable. Therefore, future changes, especially near-term changes, have made it increasingly important to be able to compute snowpack and snowmelt in sub-catchments to manage water resources.

Several studies successfully used the SWAT model to project water yield and streamflow as a function of the variable temperature and precipitation using Coupled Model Intercomparison Project Phase 3 (CMIP3 or CM3) Global Circulation Model (GCM) data sets (Ferrant et al., 2014; Shrestha et al., 2013). Neupane et al. (2014) used SWAT to simulate the effect of climate change on natural water storage at watersheds, evidencing the influence of precipitation and temperature lapse rates and inherent snow accumulation and snowmelt roles. Glacial hydrologic assessments can help track and predict water availability in catchments reliant on snowpack and timing snowmelt. The primary objective of this study is to show the scope of computation and characterization of snowpack and snowmelt in sub-catchments, which could help in understanding modeling complexities, mainly snowmelt induced in the Satluj River catchment. Another important objective of this study is to highlight the near-term future changes in snowfall and snowmelt using GCMs.

## 2. Methodology

### 2.1. Study area

The Satluj River Catchment is a part of the Indus River system, which has many major tributaries--such as the Satluj, Beas, Jhelum, Chenab and Ravi--and minor tributaries. The current research focused on a part of the Satluj River Catchment (up to Kasol gauge station), which stream flows through the western Himalayan region. The main outlet point, Kasol, consists of an area about 51055 km$^2$, which is located at the head of the Bhakra Dam of India. The geographical extent of the study area lies between 77°00' to 82°39' E longitudes and 30°8' to 33°00' N latitudes (Fig. 1). The Satluj River is the longest river among the five major rivers that flow through northern India and Pakistan. It is north of the Vindhya Range, south of the Hindu Kush segment of the Himalayas, and east of the Central Sulaiman Range in Pakistan. The Satluj Catchment is mainly covered by snow. Glaciers of the Satluj River are at moderate (526 m) to very high elevations (7429 m). The majority of the Satluj River catchment is fed by snowmelt (up to the Rampur gauge station) and rainfall during the summer and by groundwater flow during the winter.

### 2.2 Historical and near-term scenarios

Daily precipitation, minimum and maximum temperature, humidity, wind speed and solar radiation were obtained from the Indian Meteorological Department (IMD) and Indian Institute of Tropical Meteorology (IITM), Pune, India, in grid format (at 1°×1° scale). Six grids with all of these variables were kept for the drainage area of the Satluj catchment for the historical time series (1991-2008). Additionally, three gauge locations with daily measured





precipitation were also used for the same time duration. The geospatial thematic data layers such as SRTM (Shuttle
Radar Topographic Mission) digital elevation model (DEM) with 90 m spatial resolution (GLCF, 2005) and
landuse/landcover (LULC) map (prepared at 1:50,000 topographical scale using IRSP6 LISS III satellite data sets)
were used in the study. The description of the LULC codes is given in the SWAT user manual written by Neitsch et
al. (2011). A soil map of India (Figure 1) downloaded from SWAT portal (www.swat.tamu.edu/conferences/
international/2012/data set/) was also used  (FAO, 2007). The description of soil categories came from FAO's world
harmonic soil database (FAO, 2007).
For the assessment of near-term (2011 to 2030) snowpack, snowmelt and water yield, daily precipitation and
temperature data sets  were downloaded from the IPCC climate data portal. CGCM3.1/T63 atmospheric and sea-ice
model outputs–namely, SRES B2 model experiment (Qiao et al., 2013)--were used. CMIP3-SRES B2's daily
temperature and precipitation were provided at 128x64 Gaussian grid (approximately 2.81° latitudes x 2.81°
longitudes) (Thornton et al., 2009) and bias corrected ( Taylor et al., 2012; Mahmood and Babel, 2012; Singh and
Goyal, 2016b;). The SRES B2 model experiment was selected for the near-term assessment based on the
comparison of IPCC's SRES B1, SRES A2, 20C3M, COMMIT and B2 historical simulations and observed
precipitation and temperature. Four GCM data points which fell in the current study area and highlighted the spatial
variations of the present study were considered without downscaling (Fig. 1). The occurrence of snowpack and
snowmelt changes due to variations in elevation bands is enhanced by dividing the main catchment into sub-
catchments. For the sub-catchment calculations,  observed grids (six), gauge data sets (3 points) and GCM data grids
(four) were spatially interpolated at each sub-catchment using the Inverse Distance Weighting Approach (IDWA)
(Lu and Wong, 2008; Snell, 1998, Snell et al., 2000).
**2.3. Spatiotemporal approach**
Up to 10 elevation bands were incorporated in each sub-catchment to characterize the snowpack and snowmelt. For
this, at each subcatchment scale, an average TLR and PLR were computed and incorporated into the SWAT model
to improve the snowmelt and snowpack computations. The present study uses a stochastic procedure SUFI2 to
characterize model uncertainty and sensitivity analyses to improve modeling outputs in a snow-glacier dominant
Himalayan catchment. The model calibration and validation were done using daily measured discharge data at three
gauges: Rampur, Suni and Kasol. Hydro-meteorological observations, especially daily measured discharge data
from 1989 to 2008 at all three gauges, were used to improve the modeling in both historical (1991-2008) and near-
term projection scenarios (2011-2030). The historical scenarios of snowpack, snowmelt and other water balance
components were generated using hydro-meteorological data from 1991 to 2008. Near-term scenarios used two of
the most relevant GCM variables, temperature and precipitation, to produce the snowfall and snowmelt. CM3 GCM
model's daily temperature and precipitation were used to analyze the near-term complexities and changes in
snowpack and snowmelt (Ferrant et al., 2014; Shrestha et al., 2013).
The Satluj River Catchment was divided into 16 sub-catchments based on the area threshold method (Ficklin and
Barnhart, 2014; Neitsch et al., 2011). Each sub-catchment includes a main channel and multiple HRUs which





consist of geospatial representations of homogeneous land use, soil type, and management practices. The
contributions of each HRU were weight-averaged for every sub-catchment (Zhou et al., 2014). Simulated snowpack
and snowmelt were computed at each HRU and also aggregated by sub-catchment. For each sub-catchment, up to 10
elevation bands were defined; then, at each elevation band, all snow hydrological parameters were determined
(Neitsch et al., 2011). To estimate the spatial variability of the snowpack and snowmelt, an average TLR and PLR
were incorporated at each sub-catchment, which adjusted the temperature at each elevation band. Snow and glaciers
mainly cover the upper part of the catchment that has a very low TLR, while the lower part of the catchment has a
reduced presence of glacier areas with large settlements and high temperatures. These topographical variations
brought high variability in TLR and PLR over the Satluj River catchment. The rationale for discretizing the
catchment is to simulate streamflow, snowfall and snowmelt processes at each sub-catchment and the respective
elevation bands, which is also contributes to account homogeneous land use, soil, and weather generator parameters
(e.g., precipitation and temperature). A representation of the water balance components at each sub-catchment,
elevation band, and HRUs could be useful to highlight the catchments' variability in an efficient manner especially
in the case of large-area catchments.

### 147    2.3.1. Modeling approach

The SWAT model is fully capable of computing the long-term water balance components in a semi-distributed
manner through the use of hydrological response units (HRUs). Streamflow is simulated using a slope-adjusted
modified Soil Conservation Services curve number (CN) method (USDA Soil Conservation Service, 1972; Arnold
et al., 1998). Detailed physical and hydrological principles and parameters are fully described in the SWAT user
manual (Neitsch et al. 2011).

### 153    2.3.2. Model calibration and validation

Simulated and observed streamflows were used in a SWAT stochastic optimization tool, Calibration and Uncertainty
Program (CUP), to calibrate and validate physical parameters (Abbaspour et al., 2007). Recorded daily streamflows
at three outlet locations (i.e., Rampur, Suni and Kasol) for the period of 1989 to 2008 were used. The initial two
years were considered a warm-up period for the historical scenario, and the initial three years for the near-term
projected scenarios. The model calibration was performed using the concept of aggregate parameter selection (Yang
et al., 2007). An 'aggregate parameter' is obtained by adding terms such as v_, a_ and r_ to the front of the original
parameter to create an absolute increase and a relative change in the initial parameter values, respectively (Zhang et
al., 2014). The objective functions such as coefficient of determination ($R^2$) and Nash-Sutcliffe Efficiency Index
(NSE) were used in the calibration and validation procedures (Abbaspour et al., 2007; Zhang et al., 2014).

### 163    2.4. Sensitivity analysis

Model outputs are deterministic representations of precipitation, discharge, evapotranspiration (ET), storage and
different transport-processes' variables and state variables. A deterministic hydrological model such as SWAT is
unable to explore the stochastic behavior of random variables such as rainfall and associated discharges (Abbaspour





et al., 2007; Singh et al., 2013). Calibration of any distributed hydrological model using observed hydro-observation
data sets always leads to nonidentifiable parametric uncertainties due to complex hydrological processes and data
sets, especially in the case of the large catchments (Yang et al., 2007). Thus, it's necessary to estimate the
propagation of parameter uncertainty to the uncertainty of the model's outputs. SWAT uses an independent
stochastic model SUFI2, which in this work is selected to account for model uncertainty (Yang et al., 2007;
Abbaspour et al., 2007).
SUFI2 accounts for parametric uncertainty through sequential and fitting approaches. Iteratively old coefficient
parameters are updated into a new array of coefficients during calibration to ultimately achieve the final set of
parameters (Abbaspour et al. 2011). The SUFI2 algorithm assumes a large parameter uncertainty (or physically
meaningful range) occurring in response to data inputs to ensure the observed data fall into the 95% prediction
uncertainty (95PPU) band during the first iteration. During this iterative procedure, uncertainty progressive decrease
is monitored though the changes of the p-factor and r-factor  (Abbaspour et al., 2007). While the p-factor determines
the percentage of simulated data falling into the observed-data range, the r-factor contributes to determine the
uncertainty of the simulated variables and state variables when compared with observed data sets.
The value of the p-factor ranges between 0 and 100%, and the r-factor ranges between 0 and infinity. A value of p-
factor = 1 and r-factor = 0 represents a perfect match between simulated and observed data. The parameter
sensitivity analysis helps identify the significance of a particular parameter to the calibration process, whether the
process is influenced by the parameter values or nature of the forcings. SUFI2 method is based on a global
sensitivity analysis (GSA) performed through multiple regression. GSA's parameters were generated through a
Latin hypercube sampling (LHS) and the resultant simulated variables and state variables are contrasted to the
equivalent observations through the application of an objective function. The LHS method is considered a highly
efficient sampling method; it can reduce the sampling points within an individual space. In this study, four iterations
with 600 simulations were conducted to estimate the uncertain effect of model parameters in the calibration
outcomes.
In general, after completion of the first iteration, the model performed well using the majority of parameter
combinations sampled from the updated parameter ranges. Therefore, the updated parameter ranges used during the
second iteration are regarded as the uncertainty ranges for model simulations and analyses. In this study, the
statistical significance tests such as p-value test and t-stat were employed to rank parameters from high sensitive to
nonsensitive. A 0 p-values shows a highly sensitive parameter in the GSA . On the other hand, GSA's t-stat is
evaluated based on the significance level alpha ($\alpha = 0.05$) and resultant p-values. The alpha value 0.05 was chosen
as the local significance level. Based on this significance level, values larger than +1.96 indicate a significant
($p<0.05$) positive sensitivity and values lower than -1.96 indicate a significant ($p<0.05$)  negative sensitivity. Thus,
the p-values closer to zero will enable the use of trend analyses of the simulated variables and state variables
(Abbaspour et al., 2011). The parameter-sensitivity results can be observed in Table 3.
**2.5. Elevation band approach for snowpack and snowmelt measurement**





In this study, SWAT snowmelt hydrology and related processes were performed at the sub-catchment scale
(Fontaine et al., 2002). Therefore, each sub-catchment was divided into 10 elevation bands in order to incorporate
temperature and precipitation variations with respect to altitude (Neitsch et al., 2011). The sequence of
methodological steps are as follows:
**2.5.1. TLR and PLR computation and their adjustments at each elevation band**
For each sub-catchment, lapse rates for precipitation $p_{laps}$ (mm/km) and temperature $t_{laps}$ (ºC/km) were computed as
Eq. 1:
$$P_B = P + (Z_B - Z)\frac{p_{laps}}{days_{pcp,yr} \times 1000} \quad \text{and} \quad T_B = T + (Z_B - Z)\frac{t_{laps}}{1000} \tag{1}$$
where $P$ (mm), $T$ (ºC) and $Z$ (m) were the sub-catchment precipitation, temperature and recording gauge elevation,
respectively; while $P_B$, $T_B$ and $Z_B$ were the adjusted precipitation, temperature and mean elevation for each
elevation band. The variable $days_{pcp,\ yr}$ represented the mean annual number of days with precipitation. The
temperature lapse rate could be computed using mean annual temperature. In accordance with the delineation
approach used with sub-catchments, temperatures were adjusted within each elevation band by comparing the
elevation bands' midpoint elevation ($Z_B$) within the station elevation ($Z$). The elevation difference was multiplied by
the lapse rate to calculate a temperature difference between the station elevation and the elevation band. An updated
elevation band mean temperature ($T_B$) was calculated by adding or subtracting the temperature difference to or from
the temperature measured at the station elevation ($T$) as in Eq. 2:
$$T_B = T + (Z_B - Z)\frac{dT}{dZ} \tag{2}$$
where $\frac{dT}{dZ}$ is the mean local lapse rate ($t_{laps}$)(ºC/km) calculated at all sub-catchments. A lapse rate for annual
precipitation was represented by the changes of the mean annual precipitation with respect to the station elevation.
Adjusted precipitation in each elevation band ($P_B$) was based on the difference between the elevations of the
subcatchment meteorological station ($Z$) and each elevation band ($Z_B$) multiplied by the lapse rate of (mm/km) per
event ($P$). If the meteorological station was unavailable in a particular subwatershed, then the next nearest
meteorological station was considered for lapse rate calculations. The equation was defined as Eq. 3:
$$P_B = P + (Z_B - Z)\frac{dP}{dZ} \tag{3}$$
where $\frac{dP}{dZ}$ was the mean local lapse rate ($p_{laps}$) calculated for all sub-catchments.
**2.5.2 Snow Accumulation**
The snowpack was represented in SWAT by the snow water equivalent (the mass of liquid water in the snowpack)
$SWE$ (mm), which balanced snowfall $SF$ (mm) and snowmelt $SM$ (mm) or sublimation $ES$ (mm) (Eq. 4):



$$SWE_{day} = SWE_{(day-1)} + SF - SM - E_s \qquad (4)$$
In SWAT, snowmelt $SM$ is controlled by the air and snowpack temperatures, the melting rate, and areal coverage of
snow. When daily mean air temperature is less than a snowfall temperature, as specified by the SWAT variable
SFTMP (Table 1), the precipitation within an HRU is classified as snow, and the liquid water equivalent is added to
the already-present snowpack. The snowpack temperature is a function of the mean daily temperature during the
preceding days and varies as a dampened function of air temperature (Anderson, 1976). The influence of the
previous day's snowpack temperature on the current day's snowpack temperature was controlled by a lagging factor,
*(TIMP)*, which intrinsically accounts for snowpack density, snowpack depth, exposure and other factors known to
affect snowpack temperature (Eq. 5):
$$T_{snowpack(day)} = T_{snowpack(day-1)} \times (1 - TIMP) + T_{av} TIMP \qquad (5)$$
where $T_{snowpack}$ (day) and $T_{snowpack}$ (day-1) are the snowpack temperature ($°C$) on a given day and on the day
preceding it, respectively, and $T_{av}$ ($°C$) is the mean air temperature for the same given day. The fraction of area
covered by snow $SNO_{cov}$ can be computed as Eq. 6:
$$sno_{cov} = \frac{SNO}{SNO_{100}} \left( \frac{SNO}{SNO_{100}} + exp \left( cov_1 - cov_2 \frac{SNO}{SNO_{100}} \right) \right) - 1 \qquad (6)$$
where $SNO$ is the water content of the snow pack on a given day (mm), $SNO_{100}$ is the threshold depth of snow at
100% coverage (mm), and $cov_1$ and $cov_2$ are coefficients that define the shape of the curve. Snow depth over an
elevation band is assumed to be constant when the $SWE$ exceeds $SNO_{100}$; i.e., the areal depletion curve affects
snowmelt only when the snowpack water content is between zero and $SNO_{100}$.
**2.5.3. Snowmelt and glacier melt**
Snowmelt rate is controlled by snowpack temperature and air temperature. A snowpack cannot begin to melt and
release water before the entire pack has reached 0°C and thus we adopted the same. The SWAT model is unable to
calculate glacier melt contributions directly. It corresponds to snowmelt contribution mainly from the snowpack
amount. Hence, in this study the snowmelt amount integrated glacier melt and snowmelt. The melt rate from a
snowpack varies in response to snowpack conditions (Fontaine et al., 2002). In this study, snowmelt and glacier melt
were set up together in the SWAT model as a linear function of the difference between the average of the snowpack
and glacier temperature ($T_{snowpack}$) and the maximum air temperature ($T_{max}$) on a given day and the base or threshold
temperature for the snowmelt (Eq. 7). It is worth stating that due to the large number of glaciers over the Satluj
catchment, the temporal mass balance of glaciers and melting rates were analyzed at the catchment and HRU levels,
respectively. Hence, standard coefficient values were used:
$$SM = b_{mlt} \times sno_{cov} \left[ \frac{T_{snowpack} + T_{melt}}{2} - TMLT \right] \qquad (7)$$
where $b_{mlt}$ (mmH$_2$O/day-°C), is the melt factor for a day:





$$b_{mlt} = \frac{SMFMN+SMFMX}{2} + \frac{SMFMN-SMFMX}{2} sin\left(\frac{2\pi}{365}(d_n - 81)\right)$$  (8)
Eq. 8 has been adapted for application in the Northern Hemisphere, where *SMFMN* is the melt factor for 21[st] June,
*SMFMX* is the melt factor for 21[st] December, and $d_n$ represents the day of the year.

**3. Result and analysis**
Geophysical components, such as topography, land use/land cover change and soil classes, are parameterized in
SWAT (Neitsch et al., 2011) and help determine the spatial distribution of water availability and its physical-state.
For example, more than 30 different soil parameters associated with each soil category such as soil texture, available
water content, hydraulic conductivity, bulk density and organic carbon content were used for this study. These
parameters in SWAT were defined for each soil subtype for different layers (between two and three layers). A key
parameter in SWAT, the curve number (CN) averaged 80.1 in the catchment, though it varied from lower
subcatchment to upstream subcatchment as per LULC, slope and soil properties.
SWAT information on model implementation, including the temporal context for the simulated water balance
components for the sensitivity analyses is described in Table 1. The sensitivity results were a product of 20 different
hydrological parameters (Table 2) on both daily and monthly time steps. In Table 3, the parameters which were
found sensitive to snowmelt-induced streamflows are selected for model calibration. The description of parameters
and their coefficients are given in Table 3. For example, TLAPS.sub parameter (TLR) fluctuates from -7.0 ˚C/km to
2.5˚C/km (with the best-fitted value computed as -4.1 ˚C/km), showing how temperature variations exist across the
Satluj River Catchment. Table 3 also shows the aggregate parameter ranges that result from the final iteration
number, which was optimized through the Latin Hypercube Sampling (LHS) method (Abbaspour et al., 2011). For
TLAPS.sub, the p-value is recorded as 0.01 and its t-stat value is recorded as -2.2, also illustrating that this
parameter is found sensitive for the model calibration and validation.
Among the 20 calibration parameters, the 5 parameters R_CN2.mgt, R_SMFMX.bsn, V_CH_K2.rte, TLAPS.sub
and V_GW_DELAY were computed as significantly sensitive parameters for daily calibration while the parameters
SNO50COV.bsn, CN2.mgt, GW_DELAY.gw and SOL_K.sol were found sensitive for monthly analysis. The
computed  t-stat values were less than -1.96 or greater than +1.96; the estimated p-values were close to zero. At
daily time steps, sensitive parameters evidence the role of snow melt and the temperature lapse rate on water flowing
in the model. Further, soil properties also evidence the regulatory role of infiltration in the subsurface. For example,
the V_GW_DELAY.gw parameter of aquifer recharge at the catchment was found significantly sensitive for both
daily and monthly time steps (Table 3). In unconfined and shallow aquifers, this factor could influence the temporal
variability and spatial distribution of different components of the water balance, highlighting the contributions of
surface water and groundwater interactions. Also, at the catchment scale  A_ALFA_BF.gw, whose p-value was
recorded as 0.172 daily and 0.406 monthly, was found insignificant for the model calibration, indicating the
sensitivity of the model's baseflow parameterization.





Other model parameters associated with different types of LULC and soil categories were not found sensitive for the
model calibration and validation process. These model parameters included GWQMN.gw, HRU_SLP.hru,
SOL_BD.sol, HRU_SLP, PLAPS.sub, CH_N2.rte, SOL_AWC.sol and GW_REVAP. The snowmelt temperature-
related parameters such as R_SMTMP.bsn, R_SFTMP.bsn and R_SMTMP.bsn were also recorded nonsignificant
during model calibration as shown in Table 3. These properties are relevant to the temperatures that allow the
formation or accumulation of snow, rather than the melting of snow already packed (which coincides with the
sensitivity of the SMFMX parameter described above). The Saltuj River drainage area is dominated by glacial
hydrology, permanent ice sheets and seasonal well-packed snow. These, are typical features of the catchment, which
at the same time are identified in the sensitivity of parameters such as R__SNOCOVMX.bsn and SNO50COV.bsn
(parameters that represent the fraction of snowpack and the elevation bands) and the recorded significant parameter
sensitivity of daily and monthly calibration, respectively. The significance of elevation differences is reflected in the
snowpack computations. The curve number coefficient (R_CN2) was the most significantly sensitive parameter in
the model calibration process. CNs were modified based on the fractional HRU slopes so soil physical properties
could vary at sub-catchment scale. Thus, groundwater delays and baseflow, together with management practices,
soil physical properties (i.e., the CN), and snow properties, influence the generation of return flows, which aligns
with the purpose of this work in the Satluj River Catchment.
Table 4 presents the daily and monthly results for streamflow calibration (1991 to 2000) and validation (2001 to
2008) at all three outlet locations, Rampur, Suni and Kasol. Table 4 also shows the goodness-of-fit between the
simulated and measured streamflows with the coefficient of determination ($R2$) and Nash-Sutcliffe Equation (NSE)
(Legates et al., 1999) for the Rampur, Kasol and Suni outlet stations. The computed $R2$ and NSE are found
reasonably acceptable for daily and monthly observations. Regarding goodness-of-fit aspects, monthly and daily
calibration correlations were similar. Among all the three outlet stations, Kasol and Rampur show better calibration
and validation statistics than does Suni station. Before initialization of the model calibration, we took 5% as bias to
ignore the extreme ambiguities from the calibration.
Uncertainty results, which were computed using the objective functions p-factor and r-factor, provide insights about
the precision and accuracy of model simulations (Abbaspour et al., 2011). Also, factors refer to the final uncertainty
level of the calibration-validation approach. The p-factor values recorded during model calibration for the Rampur,
Kasol and Suni stations were 0.46, 0.57, and 0.52 daily and 0.41, 0.57, and 0.49 monthly for the timespan 1991-
2000 (Table 4). During model validation, the p-factor values recorded were 0.43, 0.52, and 0.53 dailyand 0.45, 0.60
and 0.58 monthly. Along the Satluj River Catchment, resultant p-factors indicate that more than 50% of the
simulated flows were encompassed within the uncertainty bonds for Kasol station's daily and monthly simulations,
as well as for calibration and validation approaches. In contrast, simulated flows for Rampur showed p-factors
below 50%, contrasting with their performance on the SWAT model for Kasol and Suni stations during daily
simulations and for model validation. On the other hand, the r-factor values recorded were 1.89, 1.50, and 1.60 daily
and 1.90, 1.57, and1.43 monthly for Rampur, Kasol and Suni. During model validation, the r-factor values were
calculated as 1.89, 1.67, and 1.72 daily and 1.92, 1.62 and 1.52 monthly for Rampur, Kasol and Suni. Resultant r-





factors indicated the SWAT's ability to precisely reproduce flow values; however, values above 1.43 indicated that other sources of error besides model physics could contribute to the values of the r-factor. The experiments described here are unable to identify the contribution of such sources of error. Kasol, Suni, and Rampur were the only stations with observed data and all were located in the lowest drainage area in the Sultej River Catchment. Although small, differences among model performance metrics illustrate the local contributions of Suni and Rampur's downstream drainage areas to the total streamflow generated at Kasol. Kasol "averages" over and under estimations of streamflows generated upstream, so lower r-factor values are expected, representing higher precision. Further, smaller values of p-factor in Kasol also indicate a lower accuracy of the model in replicating observed streamflows within the uncertainty bonds.

Considering that most of the drainage areas of this catchment are snowmelt-dependent and are upstream of Rampur station, a deeper assessment of snowfall and snowmelt along with streamflow generation is required at high altitudes. The temporal variability and spatial distribution of the hydrological components such as precipitation, snowpack, snowmelt, water yield (contributed by rainfall only) and total water yield (contributed by both snowmelt and rainfall) were computed and analyzed. Figure 2 illustrates the aggregation of simulated snowpack and snowmelt compared with precipitation from 1991 to 2008 in sub-catchments. Here, it is evident that the maximum snowpack contribution occurs at sub-catchments at a high elevation. These sub-catchments, such as SB1, SB2, SB3, SB4, SB15 and SB16, have values varying from ~10 to ~380 mm in a single year. Figure 2 also shows that sub-catchments such as SB10, SB11, SB12, and SB13, located in the lowest drainage areas, poorly contribute to the snowpack of the Saltej River Catchment. (They had annual values below 150 mm predominantly). Interannual changes in snowpack and the amount precipitation show local to large-scale influences in snow melt as well as snow accumulation. For example, SB1 shows that the proportion of snowmelt/snowpack with respect to precipitation was larger in 2000 and 2002, which contrasts with those proportions between 1995 and 1996. In the easternmost portion of the catchment, this proportion is consistent during all years, which contrasts with the catchment's lower drainage areas. Further analysis is required to identify causality in those accumulations in response to El Niño Southern Oscillation or interannual changes in monsoon intensity and interannual accumulation of snow. Figure 3 illustrates a possible influence of elevation differences along the catchment.

During near-term projection, input parameters such as DEM, LULC, and soil map were kept constant to simulate and isolate possible effects of temperature and precipitation, which could emerge in places with highly variable elevations and large elevation gradients. The TLR and PLR were estimated by elevation band as shown in Figure 3. The TLR and PLR are given as an input to set up the SWAT model for sub-catchment calculations of snowfall and snowmelt, as well as parameters in calibration. Figures 3a and 3b illustrate the TLR or inverse changes in temperature with altitude (Gardner et al., 2009). Figures 3a and 3b also show the winter and summer months' temperature variations in relation to elevation differences, as well as the inherent variation due to seasonal cycles at each sub-catchment. While winter temperatures in low-altitude portions of the catchment vary between 9°C and 21°C, summer temperatures range between 22°C and 27°C. At high altitudes, the largest temperature span (21°C) occurs during winter months whereas the summer months' temperature span (5°C) remains the same along the





catchment. Parameter sensitivity in daily and monthly analyses (described in Table 3) evidenced SWAT's ability to
simulate flows in response to snowmelt rather than changes in temperatures. Figure 3b evidences such sensitivity
since the temperature between April and September remains within a $5^{\circ}$C temperature span.
Figure 4 shows annual averages of snowpack variations by elevation band (10 numbers) computed at each sub-
catchment for the 1991-2030 period. These variations are expressed in fractional snowpack at each sub-catchment,
which at the same time define the variations in TLRs and PLRs. The distribution of the fractional snowpack varied
throughout the catchment from upstream to downstream sub-catchments. Figures 4a to 4d are examples of high-
altitude drainage areas characterized by high and variable snowpacks. In contrast, low-land variations upstream of
Rampur station (Figures 4e and 4f) evidenced small variability and low values of accumulated snow. Downstream of
Rampur (Figures 4g and 4h) illustrate slightly larger variations in snow accumulation with average values below 50
mm/year. Figure 5 is consistent with the fractional variations in snowpack expressed above, expanding such
variations into multidecade contributions (1991-2000, 2001-2008, 2011-2020 and 2021-2030). In this figure,
snowpack variation is highlighted at each catchment on a cumulative annual average. Figure 5 shows that sub-
catchments at high elevations, such as SB1, SB2, SB3, SB8, SB15 and SB16, receive the highest amounts of
snowpack. When compared intra-annually, the scenarios computed between 1991-2000 and 2001-2008 showed
higher snowpack amounts than those calculated between 2011-2020 and 2021-2030. This difference in snowpack
amount mainly occurred due to the variations in fractional snow covers.
Figures 6a-e show the spatial distribution of multidecadal averages of precipitation, snowpack, snowmelt, rainfall-
runoff and total water yields (contributed by both snowmelt runoff and rainfall runoff) for the period 1991-2008 and
their differences with respect to the near-term period 2011-2030. Figure 6a shows that the lower portion of the
catchment (i.e., SB10, SB11, SB12 and SB13) and highest elevated part of the catchment (i.e., SB14, SB15 and
SB16) had the largest precipitation (1991-2030). However, when compared with split time series sets, such as the
1991-2008 and 2011-2030 time series sets, precipitation decreases in the high elevation sub-catchments and
increases in the lower parts. The snowpack and snowmelt plots have shown similar kinds of trends in their time
series values. A decrease in the snowpack amounts can be observed in Figures 6b and 6c. Figure 6d also shows that
the contribution of runoff (due to rainfall) has increased during the time 2011-2030. Figure 6e shows an increase in
total water yield in subcatchments at low elevations. The portions of the watershed most vulnerable to hydrologic
changes, specifically responses to variations in snow melting and snow accumulation, are the mid- to low-altitude
portions of the catchment upstream of Rampur station.
Figure 7 illustrates the magnitude of change (shown as "% of change") in snowpack amount as a function of the
fraction of elevation bands. The results showed a decrease in snowpack amount recorded from a minimum of 5% to
a maximum of 42% across all the subcatchments. The subcatchments SB1, SB2, SB3 and SB8 correspond with the
utmost decrease in snowpack amount (20% to 42%); whereas, the subcatchments SB5, SB7, SB14, SB15 and SB16
showed a small to moderate decrease in snowpack amount (4% to 20%). The above showed significant variations in
the water balance components of the Satluj River catchment, illustrating an enormous change in snowpack amount
over different sub-catchments.





## 5. Conclusion

This study analyzed the snowpack and snowmelt computations in high elevations of the Satluj River Himalayan catchment. In this study, the snowpack and snowmelt have been evaluated at multiple elevation bands, illustrating spatial variations in their amount at each subcatchment. For the computation of snowpack and snowmelt, both measured and GCM data sets were used to highlight the intraannual changes in snowmelt and snowpack. This study showed an enormous spatial and temporal variability in snowpack amount at elevation bands. The average TLR and PLR were used to compute the more accurate estimation of snowpack. For this, various model calibration parameters were considered and then sensitivity was analyzed. Based on the sensitivity analysis, significant sensitive and nonsensitive parameters were identified, which helped to improve the accuracy of the computation of snowpack and snowmelt. The other water balance components such as precipitation, water yield due to rainfall and water yield due to snowmelt were spatial studies. The long-term spatial comparison of these water balance components showed noticeable spatial variability from upstream subcatchments to downstream subcatchments. The percentage of change analysis clearly showed that snowpack is highly variable over the Satluj catchment and it could be more variable in the near-term period.

**Acknowledgment**

We sincerely thank the India-WRIS project (RRSC-W, Indian Space Research Organization, India) and Central Water Commission (New Delhi, India) for providing the necessary data to successfully complete this research. We also thank the Intergovernmental Panel on Climate Change (IPCC) for providing the necessary GCM data sets for analysis.

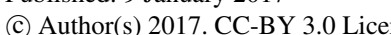



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



**Tables**
Table 1: Details of water balance components simulated in the SWAT model

| Model Simulation Details | |
| --- | --- |
| General details | Satluj catchment |
| Simulation period (years) | 16 |
| Warmup (years) | 3 |
| Hydrological response units | 358 |
| Sub-catchments | 16 |
| Output time step | Daily, Monthly |
| Watershed area (km$^2$) | 51055 |
| Water Balance Ratios | |
| Streamflow/precipitation | 0.63 |
| Baseflow/total flow | 0.25 |
| Surface runoff/total flow | 0.45 |
| Percolation/precipitation | 0.26 |
| Deep recharge/precipitation | 0.01 |
| ET/precipitation | 0.36 |
| Water Balance Components (mm) | |
| ET | 382.0 |
| Precipitation | 1073.5 |
| Surface runoff | 304.8 |
| Lateral flow | 113.0 |
| Return flow | 259.0 |
| Percolation to shallow aquifer | 283.4 |
| Revaporation from shallow aquifer | 10.2 |
| Recharge to deep aquifer | 14.2 |











Table 2: Description of model calibration parameters

| Streamflow parameters selected for calibration and validation | Description |
| --- | --- |
| SNOCOVMX.bsn | Minimum snow water content |
| HRU_SLP.hru | Average slope steepness |
| SOL_K.sol | Soil hydraulic conductivity |
| SNO50COV.bsn | Fraction of snow volume |
| PLAPS.sub | Precipitation lapse rate |
| SFTMP.bsn | Snowfall temperature |
| GWQMN.gw | Threshold depth of water in shallow aquifer required for return flow |
| CH_N2.rte | Manning roughness coefficient for main channel |
| SOL_BD.sol | Moist bulk density |
| SOL_AWC.sol | Available water capacity of the soil layer |
| GW_REVAP.gw | Groundwater "revaporation" coefficient |
| SMTMP.bsn | Snowmelt base temperature |
| ALPHA_BF.gw | Baseflow alfa factor coefficient |
| SMFMN.bsn | Melt factor for snow on December 21[st] |
| SOL_Z.sol | Depth from soil surface to bottom layer |
| GW_DELAY.gw | Groundwater delay time |
| TLAPS.sub | Temperature lapse rate |
| CH_K2.rte | Effective hydraulic conductivity |
| SMFMX.bsn | Melt factor for snow on June 21[st] |
| CN2.mgt | Curve number coefficient |












Table 3: Aggregate parameters and their values, ranges and global sensitivity results

| SI. No. | Parameter | Daily | | | | |
| | | Fitted Value | Minimum Value | Maximum Value | t-Stat | P-Value |
|---|---|---|---|---|---|---|
| 1 | A__SNOCOVMX.bsn | 300.0 | 0.0 | 500.0 | -2.1 | 0.03 |
| 2 | R__HRU_SLP.hru | 0.2 | 0.2 | 0.2 | -0.2 | 0.9 |
| 3 | R__SOL_K.sol | 0.3 | 0.0 | 1.3 | -0.2 | 0.8 |
| 4 | R__SNO50COV.bsn | 0.4 | 0.0 | 50.0 | -0.2 | 0.8 |
| 5 | A__PLAPS.sub | 277.0 | 100.0 | 300.0 | 0.3 | 0.8 |
| 6 | A__SFTMP.bsn | -1.7 | -1.8 | 1.0 | -0.4 | 0.7 |
| 7 | V__GWQMN.gw | 1.0 | 0.8 | 1.1 | -0.4 | 0.7 |
| 8 | V__CH_N2.rte | 0.3 | 0.2 | 0.3 | 0.5 | 0.6 |
| 9 | R__SOL_BD.sol | 1.4 | 1.2 | 1.5 | 0.7 | 0.5 |
| 10 | R__SOL_AWC.sol | 0.6 | 0.6 | 0.7 | 0.7 | 0.5 |
| 11 | V__GW_REVAP.gw | 0.0 | 0.0 | 0.0 | 0.9 | 0.4 |
| 12 | R__SMTMP.bsn | -0.5 | -2.7 | 2.0 | -0.9 | 0.4 |
| 13 | A__ALPHA_BF.gw | 0.12 | 0.06 | 0.2 | -1.4 | 0.2 |
| 14 | R__SMFMN.bsn | 7.4 | 6.4 | 7.7 | -1.4 | 0.2 |
| 15 | R__SOL_Z.sol | 2813 | 100 | 4000.0 | 1.6 | 0.1 |
| 16 | V__GW_DELAY.gw | 10.5 | -88.6 | 50.1 | -2.1 | 0.02 |





| | | | | | | |
|---|---|---|---|---|---|---|
| 17 | A__TLAPS.sub | -4.1 | -7.0 | 2.5 | -2.2 | 0.01 |
| 18 | V__CH_K2.rte | 27 | 22 | 75.0 | 2.4 | 0.0 |
| 19 | R__SMFMX.bsn | 0.5 | -0.5 | 1.4 | 6.4 | 0.0 |
| 20 | R__CN2.mgt | 0.03 | 0.0 | 0.1 | -8.6 | 0.0 |
| Monthly | | | | | | |
| 1 | R__SOL_BD.sol | 1.0 | 0.9 | 1.6 | -0.1 | 0.9 |
| 2 | R__SMFMN.bsn | 9.1 | 6.2 | 11.2 | 0.3 | 0.8 |
| 3 | A__PLAPS.sub | 337 | 100 | 350.0 | -0.3 | 0.7 |
| 4 | V__GW_REVAP.gw | 0.1 | 0.1 | 0.2 | -0.4 | 0.7 |
| 5 | R__TLAPS.sub | -4.6 | -6.2 | 2.5 | 0.4 | 0.7 |
| 6 | V__GWQMN.gw | 1.6 | 0.8 | 1.7 | 0.7 | 0.5 |
| 7 | R__SOL_AWC.sol | 0.5 | 0.5 | 0.9 | -0.8 | 0.4 |
| 8 | R__SOL_Z.sol | 1296 | 1265 | 4388.0 | -0.8 | 0.4 |
| 9 | V__ALPHA_BF.gw | 0.11 | 0.0 | 1.7 | 0.8 | 0.4 |
| 10 | R__SNOCOVMX.bsn | 100 | 50 | 500.0 | -1.0 | 0.3 |
| 11 | R__SMTMP.bsn | 0.7 | 0.6 | 1.7 | 1.1 | 0.3 |
| 12 | R__SFTMP.bsn | 1.4 | 1.0 | 1.9 | 1.4 | 0.2 |
| 13 | R__SMFMX.bsn | 0.4 | 0.3 | 1.5 | -1.8 | 0.1 |





| | | | | | | |
|---|---|---|---|---|---|---|
| 14 | R__SOL_K.sol | 0.5 | 0.6 | 1.3 | 2.1 | 0.0 |
| 15 | V__GW_DELAY.gw | 20 | -70 | 251 | 2.6 | 0.0 |
| 16 | R__CN2.mgt | 0.02 | 0.0 | 0.1 | -4.6 | 0.0 |
| 17 | R__SNO50COV.bsn | 0.2 | 0.0 | 1.0 | -11.7 | 0.0 |

















Table 4: Model calibration and validation results using the SUFI method for the daily and monthly
analysis

| Outlet Station | Calibration (1991 - 2000) | | | | | |
| | Daily | | | Monthly | | |
| | p-factor | r-factor | $R^2$ | p-factor | r-factor | $R^2$ |
|---|---|---|---|---|---|---|
| Rampur | 0.46 | 1.89 | 0.75 | 0.41 | 1.90 | 0.71 |
| Kasol | 0.57 | 1.50 | 0.76 | 0.57 | 1.57 | 0.78 |
| Suni | 0.52 | 1.60 | 0.72 | 0.49 | 1.43 | 0.73 |
| Outlet Station | Validation (2001 - 2008) | | | | | |
| | Daily | | | Monthly | | |
| | p-factor | r-factor | $R^2$ | p-factor | r-factor | $R^2$ |
| Rampur | 0.43 | 1.89 | 0.62 | 0.45 | 1.92 | 0.65 |
| Kasol | 0.52 | 1.67 | 0.71 | 0.60 | 1.62 | 0.73 |
| Suni | 0.52 | 1.72 | 0.65 | 0.58 | 1.52 | 0.71 |



















**Figure Captions**
**Fig. 1**: Study area map of Satluj river catchment (up to Kasol station/gauge).
**Fig. 2**: Sub-catchment and annual variability in snowpack and snowmelt (annual average) for the year
1991 to 2008.
**Fig. 3**: Distribution of average temperature over the sub-watershed's centroid elevation (in chronological
order); (a) winter season and (b) summer season.
**Figure 4**: (a-h) Sub-catchment snowpack variability (average annual) based on the fractional elevation
bands in long term climate domain (1991-2030) and (b) Cumulative variability in snowpack amount over
different sub-catchments of Satluj catchment in different temporal domains.
**Figure 5**: Cumulative variability in snowpack amount over different sub-catchments of Satluj catchment
in different temporal domains.
**Fig. 6**: Historical average (1991-2008) and differences between near-term and historical average for (a
and b) precipitation, (c and d) snowpack/snowfall, (e and f) snowmelt, (g and h) water yield (due to snow)
and (I and j) total water yield (snowmelt and rainfall runoff) in the Satluj River Basin.
**Fig. 7**: Percentage of change in snowpack amount (average annual) over different sub-catchments of
Satluj River.










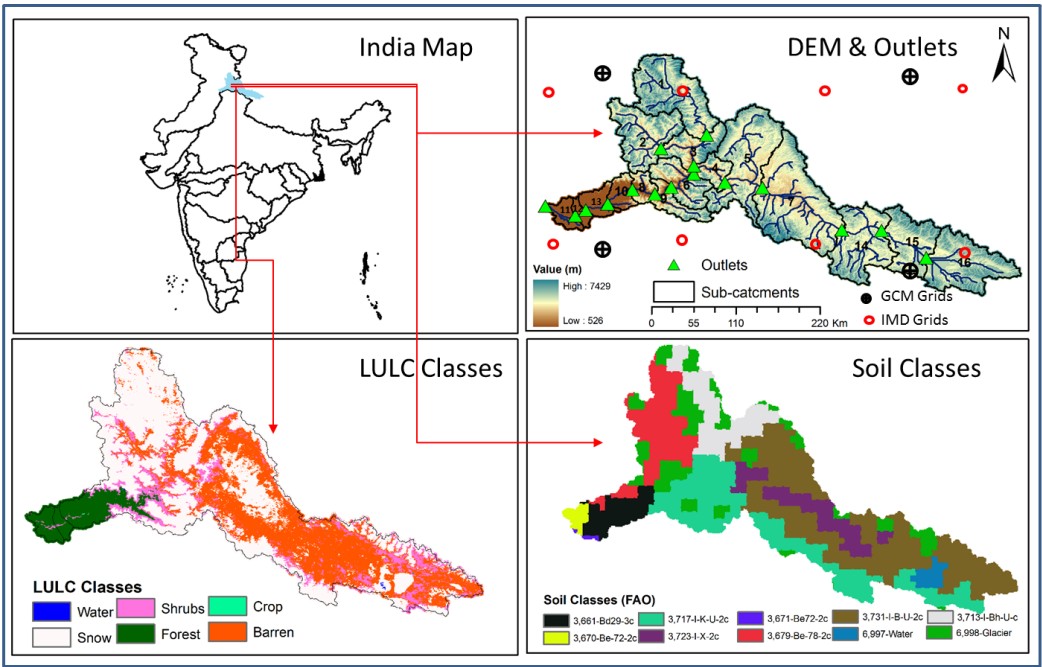

**Fig. 1**: Study area map of Satluj river catchment (up to Kasol station/gauge).

















**Fig. 2**: Sub-catchment and annual variability in snowpack and snowmelt (annual average) for the year
1991 to 2008.





(a)

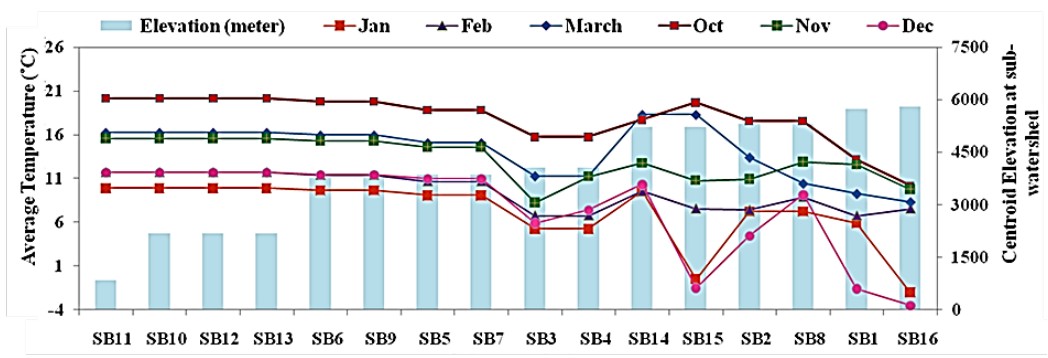


(b)

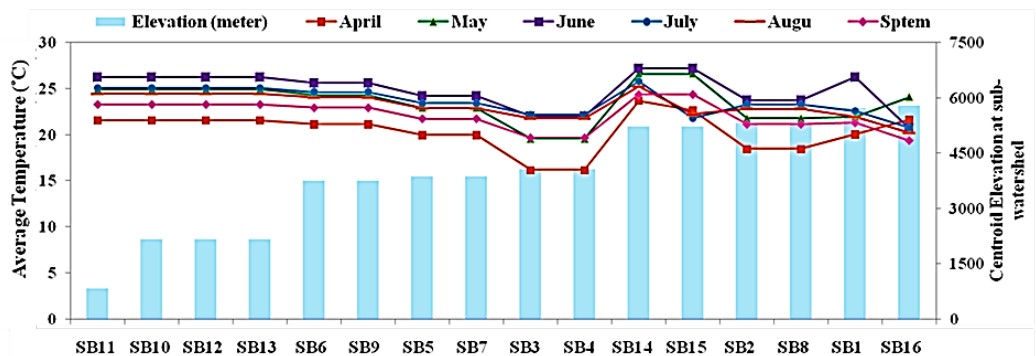


**Fig. 3**: Distribution of average temperature over the sub-watershed's centroid elevation (in chronological
order); (a) winter season and (b) summer season.









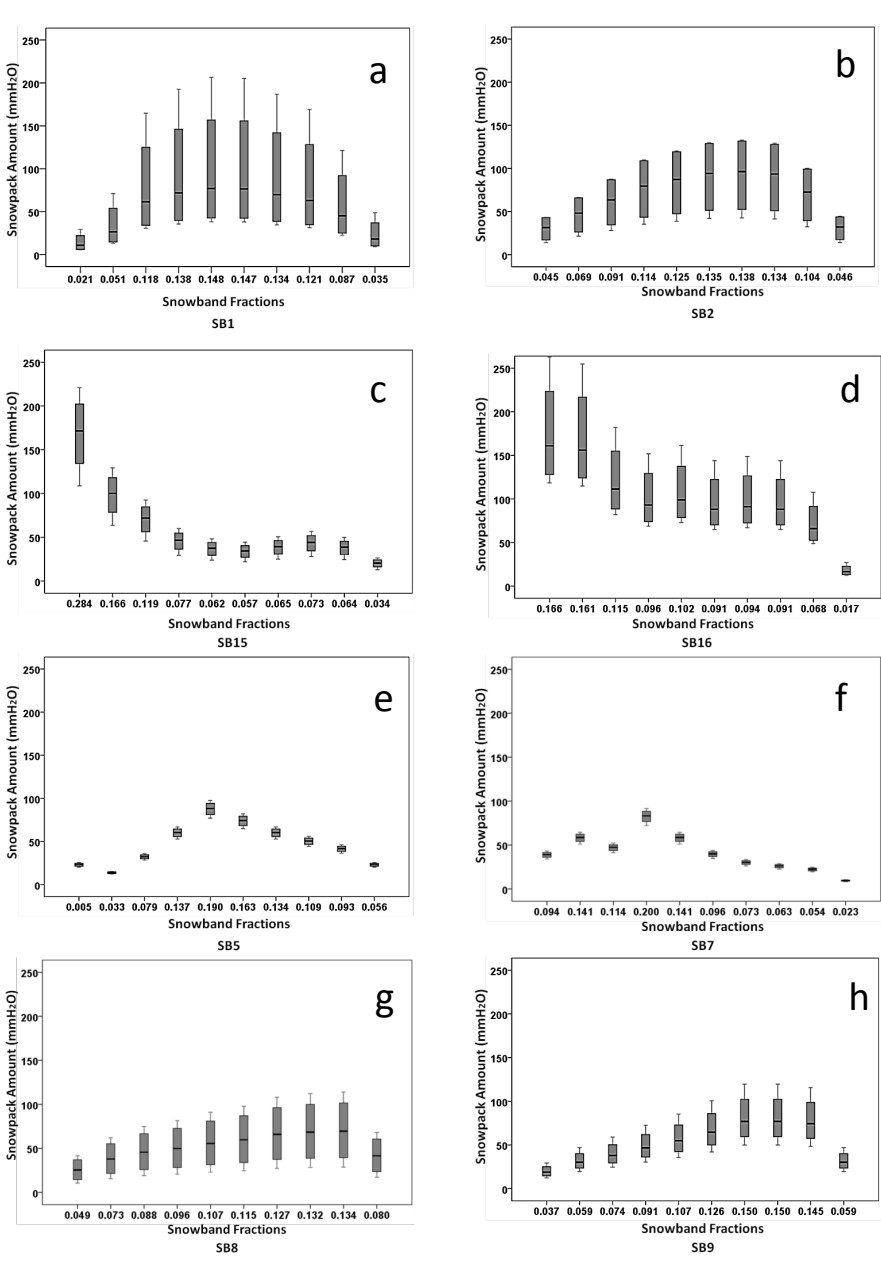


**Figure 4**: (a-h) Sub-catchment snowpack variability (average annual) based on the fractional elevation
bands in long term climate domain (1991-2030) and (b) Cumulative variability in snowpack amount over
different sub-catchments of Satluj catchment in different temporal domains.






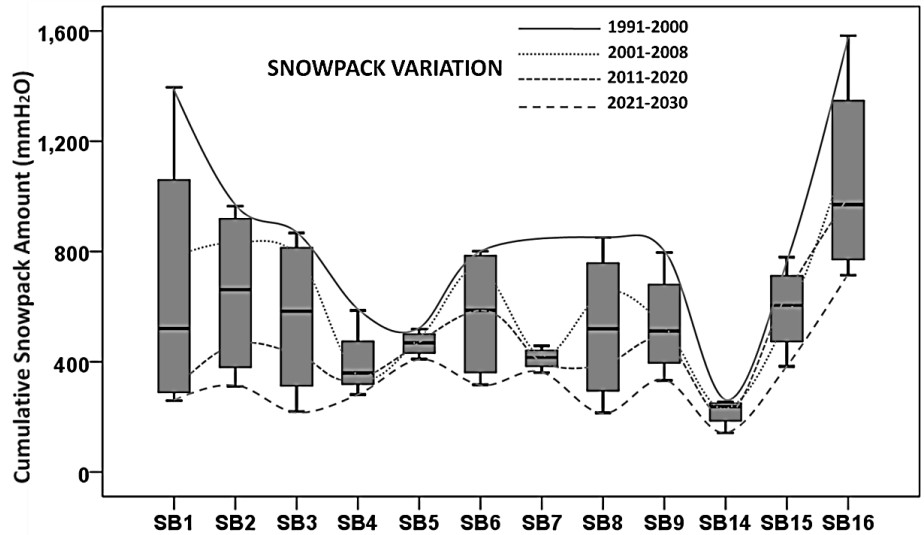


**Figure 5**: Cumulative variability in snowpack amount over different sub-catchments of Satluj catchment

in different temporal domains.



















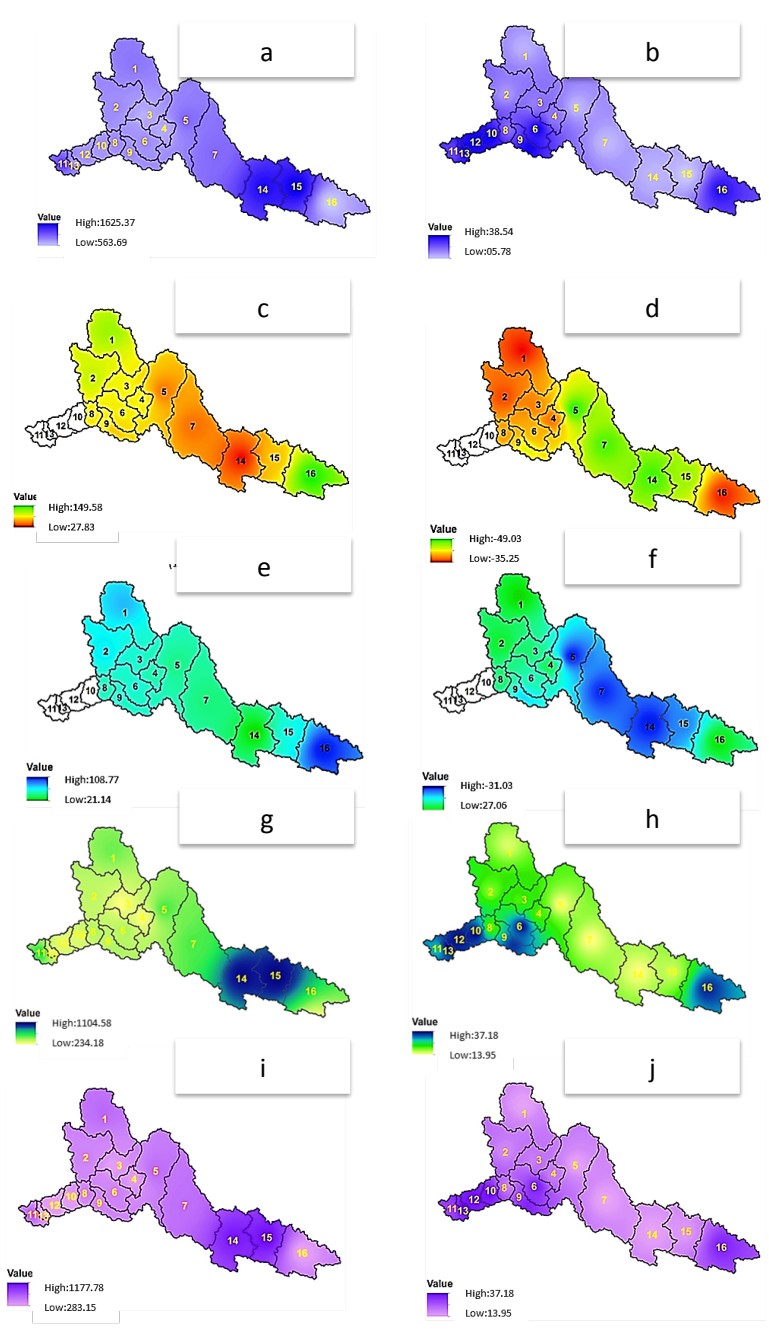


**Fig. 6**: Historical average (1991-2008) and differences between near-term and historical average for (a and b) precipitation, (c and d) snowpack/snowfall, (e and f) snowmelt, (g and h) water yield (due to snow) and (I and j) total water yield (snowmelt and rainfall runoff) in the Satluj River Basin.





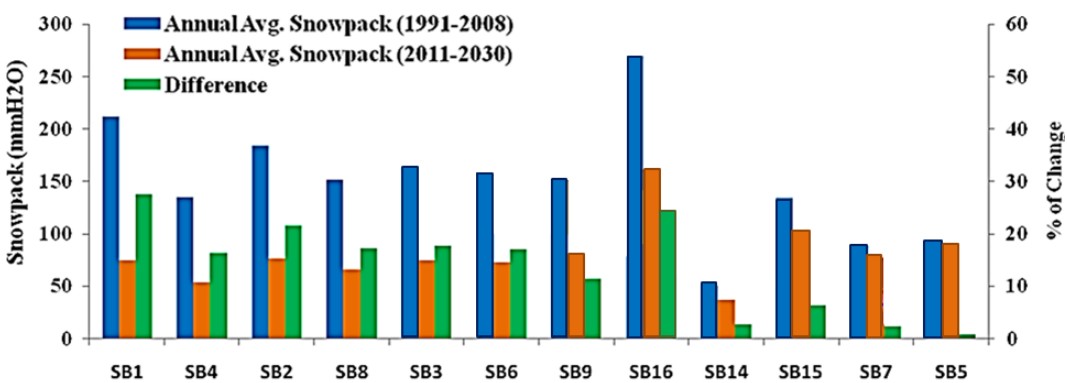


**Fig. 7**: Percentage of change in snowpack amount (average annual) over different sub-catchments of
Satluj River.
