# Peer review of "Sub catchment Assessment of snowpack and snowmelt change"

_Hydrology and Earth System Sciences, 2016_

## Referee Comment (RC1) · Anonymous Referee #1 · 29 Jan 2017

Summary

This submission is a typical example of a modeling study that is using poor input data and large assumptions to make great inferences about the hydrological response of a watershed. The study is attempting to assess the role of elevation on snow accumulation and snowmelt contributions to streamflow in a data-poor river basin in the Himalayas. The authors provide very little information on how spatial units are discretized and how snow accumulation is estimated from CMIP-3 temperature and precipitation data for the 51,000 km2 basin that has one of the largest variability in elevation and

relief found on this planet. The study describes how elevation bands are derived and mentions the use of meteorological station data to estimate the temperature lapse rate, however, the methods section does not contain any information on how many meteorological stations were included in the analysis nor where these stations are located. Similarly, the accuracy of the simulated snowfall is not validated with snow cover data or on-the-ground measurements making this input dataset highly questionable. The 'simulated' snowfall is then used as an input dataset to the SWAT model to predict the change in snowmelt runoff until 2030. The study does not even show a graph of the observed versus modeled streamflow to indicate the goodness of fit of the model (goodness of fit measures are only provided in the form of a table) thus it is highly questionable that the model is capturing runoff dynamics (e.g. peak flows, base flows) correctly. This study would have much more scientific value if the authors would attempt to improve uncertainty propagation of these huge uncertainties in the input data. Beyond these coarse assumptions, the study is also somewhat limited in scope in such that it only considers changes in snowmelt and largely ignores melt water contributions from glacier melt. As the authors clearly point out in their introduction, the Himalayas undergo drastic change in both snowmelt and glacier melt. Forward prediction of the snowmelt dynamics is therefore of limited value if the model does not include energy balance couplings to glacier melt. The authors therefore should clearly state up-front (e.g. in their abstract and study objectives) that the analysis has been done assuming that there is limited change in glacier melt over the time period considered or develop a glacier melt algorithm for the SWAT model.

Specific comments

Line 19: Suggest adding the size of the watershed considered in this study. This info is particularly of interest to assess how representative this study is for the entire Himalayan region.

Line 26: Delete "approach"

Line 25 ff.: It is not clear from the abstract what the SWAT model is used for? The abstract could be streamlined to clearly state what the objective of the study is and what the different datasets and tools (e.g. SWAT) are used for. Also it would be good to clarify what snowmelt module was used in SWAT.

Line 74: "timing snowmelt"? Timing of snowmelt? How is glacier melt considered in the SWAT model? How is the model calibrated to consider seasonal changes in glacier mass balance, which includes quite a bit of snow?

Line 76: Wording! "understanding modeling complexities, mainly snowmelt induced..."? Do you refer to snowmelt induced modeling complexities? Please revise!

Line 84: Delete "stream" in front of flows.

Line 85: What percentage of the entire Himalayan region does the Satluj river basin represent?

Line 89: What is the percentage of glacier cover in each of the 14 sub-catchments?

Line 110: What does SRES B2 stand for? This is not the most common way emission scenarios are referred to. Also it would be valuable to state what this scenario's main assumptions are.

Line 120: Can you clearly state how many meteorological stations and snow monitoring stations were included in this analysis and where these stations are located? Right now the manuscript suggests that the SWAT model uses only IPCC temperature and precipitation data, is then modeling the snowfall and snow accumulation and snowmelt. How much effort was spent to validate the quality and validity of the IPCC products in this river basin? There have been numerous studies concluding that snow accumulation in mountainous regions is highly variable. Without this uncertainty analysis in the input data the SUFI2 sensitivity analysis is worthless.

Line 126: How were historical scenarios of snowpack, snowmelt and other water balance components generated?

Line 131: The abstract states (line 21) that the basin was divided into 14 sub-catchment. Here you state 16. Which one is correct?

Line 134: A little bit more information on the HRU delineation is needed. How many HRUs were derived for each sub-catchment? What curve numbers were assumed for the different soil types (which are not clearly defined in Figure 1!)? How were the 10 elevation bands derived (e.g. natural break, equal interval, equal area)? Why not use local information on the lapse rate to define the elevation bands?

Line 140: More information on glacier cover in each sub-catchment is needed.

Line 181: The SUFI2 approach might be useful for a watershed where the modeler has high confidence in the input data. Given that this study is using FAO soil maps (1x1 degree), IPPC gridded temperature and precipitation data which aggregate rainfall and snowfall input over a large region that does not take any spatial difference due to topography into account, the approach use of the SUFI2 is useless in the uncertainty in the input data is not properly addressed. This study would have much more scientific value if the authors would attempt to improve uncertainty propagation of these huge uncertainties in the input data.

Line 216: It is not clear what stations were used to calculate the temperature difference and lapse rate for the elevation bands. The methods section does not contain any information on how many meteorological stations were included in the analysis.

Line 252: The authors state clearly that SWAT is not capable to simulate glacier melt. However, 5 lines further down they claim that glacier melt was integrated with the snowmelt. Glacier melt presents a streamflow contribution from a storage reservoir, which cannot or should not be captured by the snowmelt routine. Once the accumulated precipitation is exhausted no more runoff can be contributed by snowmelt, however, this is typically the time when glacier melt contributions dominate. The authors

state that there is a large number of glaciers in the basin (this should be clearly stated in the methods and basin description!) which can contribute glacier melt. Equation 7 is not capable to adequately capture glacier melt once the snow cover (snocov) is zero.

Line 302: "The Saltuj River drainage area is dominated by glacial hydrology, permanent ice sheets and seasonal well-packed snow." This sentence captures nicely the variability in snow and ice cover in the basin and illustrates the spatial and temporal variability of snowmelt processes that prevail in the basin, which cannot be captured by a melt rate model. Well-packed snow has a much greater density that freshly fallen snow and melts differently that new snow, aged snow or glacier ice. Since the authors do not provide any information on the HRU discretization of each sub-watershed I assume these spatial differences in snow and melt patterns are not accurately captured by the model.

Line 394: The finding that low- to mid-latitude regions experience the most change from snowmelt to rainfall runoff is not surprising has been found in many other mountainous regions across the world (Foster et al. 2016, ERL; Klos et al. 2014 GRL).

Line 397: How would a change in elevation bands influence the results shown in Figure 7?

Figure 1: What do the black circles with the cross mean? Please ensure that each map shows a complete legend! The legend of the soil map is very uninformative. Could the different soils be grouped into know soil types or soil families? The 16 sub-catchments and their boundaries are hard to see. Please use a different line and text color for the catchment boundaries. Please add locations of glaciers within the basin. Also the locations of the three gauges mentioned in the methods cannot be found in any of the figures.

Figure 2: Please correct the spelling of "snowmelt" in each graph. Please define what you mean by snowpack and snowmelt. Is snowpack the depth of the snowpack and snowmelt the snow water equivalent?

Figure 4: The figure caption for this figure does not make sense. Graph (b) does not seem to show the cumulative variability in snowpack amount as stated in the caption.

Figure 6: This figure is somewhat awkward. The labels for each plot are too big and cover in some instances the map. I would suggest putting each map into a grid/frame that is then clearly labeled with both a-j and the thematic title of the map.

Fig. 7: Why are the bar graphs not sorted in increasing order from SB1-SB16 along the x-axis? SB5 is the last bar with the least change in snowpack but it is not clear what elevation band this sub-catchment belongs to.

---

## Referee Comment (RC2) · Anonymous Referee #2 · 6 Feb 2017

General comments

The study aims to improve simulation of snowmelt runoff generation in Himalayas and to estimate its future changes.

The topic is interesting and relevant from both – scientific and water resources management perspective. The manuscript in its current form, however, does not clearly formulate the novel scientific contribution. There are numerous studies presenting variability and change in snow accumulation and melt in Himalayas, including future projections, however it is not clear in which aspects is this study new. The calibration of

a model and applying it then for simulations of future projections is by no means novel. The discussion of the results is missing, but this would be an opportunity to clearly show the contribution of the study with respect to existing results. More importantly, the justification of the model selection (used in the study) is not clear. I wonder how the SWAT model (CN approach) represents the dominant runoff processes in Himalayas. How does it account for glacier melt, snow accumulation and melt on steep slopes, the snow drift, energy balance, etc (i.e. processes and their changes related to snow melt generation)? Is the surface runoff really contributing more than 40% of annual runoff (table 1) in such large basin? I really missed to see some time-series (observed versus simulated) and process-based interpretation (with justification) of results.

I'm sorry, but I do not recommend to publish the study in its current form. At least a significant revision is needed.

---

## Author Comment (AC1) · 4 May 2017

Reply from Authors Reviewer# 1 This submission is a typical example of a modeling study that is using poor input data and large assumptions to make great inferences about the hydrological response of a watershed. The study is attempting to assess the role of elevation on snow accumulation and snowmelt contributions to streamflow in a data-poor river basin in the Himalayas. The authors provide very little information on how spatial units are discretized and how snow accumulation is estimated from CMIP-3 temperature and precipitation data for the 51,000 km2 basin that has one of

the largest variability in elevation and relief found on this planet. The study describes how elevation bands are derived and mentions the use of meteorological station data to estimate the temperature lapse rate, however, the methods section does not contain any information on how many meteorological stations were included in the analysis nor where these stations are located. Similarly, the accuracy of the simulated snowfall is not validated with snow cover data or on-the-ground measurements making this input dataset highly questionable. The 'simulated' snowfall is then used as an input dataset to the SWAT model to predict the change in snowmelt runoff until 2030. The study does not even show a graph of the observed versus modeled streamflow to indicate the goodness of fit of the model (goodness of fit measures are only provided in the form of a table) thus it is highly questionable that the model is capturing runoff dynamics (e.g. peak flows, base flows) correctly. This study would have much more scientific value if the authors would attempt to improve uncertainty propagation of these huge uncertainties in the input data. Beyond these coarse assumptions, the study is also somewhat limited in scope in such that it only considers changes in snowmelt and largely ignores melt water contributions from glacier melt. As the authors clearly point out in their introduction, the Himalayas undergo drastic change in both snowmelt and glacier melt. Forward prediction of the snowmelt dynamics is therefore of limited value if the model does not include energy balance couplings to glacier melt. The authors therefore should clearly state up-front (e.g. in their abstract and study objectives) that the analysis has been done assuming that there is limited change in glacier melt over the time period considered or develop a glacier melt algorithm for the SWAT model.

We appreciate the valuable comments made by reviewer 1. We address those comments integrating the reviwer-1's suggestions, as well as the manuscript scope, data availability and analyzes proposed. Below you will see the statements and/or paragraphs written, as examples of how we address Reviewer-1 comments in four sections based on the general review as follows:

1- ". . . however, the methods section does not contain any information on how many

meteorological stations were included in the analysis nor where these stations are located."

Additional details and clarity have been added to input data descriptions, which led to an enhanced narrative connected with updated Figures (See Figure 1) illustrating the location of the stations.

2- "the accuracy of the simulated snowfall is not validated with snow cover data or on-the-ground measurements making this input dataset highly questionable"

In this paper, we performed a stochastic calibration using observed streamflows, and validated the snow-melt induced streamflows at three gauges. The relevant snowmelt parameters are taken into consideration during model setup and calibration. In this paper, we did not focus on the validation of snowpack estimates using real time satellite datasets. Also, because field observations were absent, this study based their assessment on how we can compute accurate snowpack and snowmelt changes in response to changes in altitude. We supported the analyses with standard coefficients, previously reported/validated on literature, relevant to snowmelt hydrology for the computation of snowmelt and snowpack up to 2030 (and NOT 2130 as we put in the text by mistake). The temperature index model utilizes observed meteorological datasets such as precipitation and temperature and thus snowpack and snowmelt computation has been expected as equivalent to satellite-based snow covers. Apart from this, we also utilized temperature lapse rate and precipitation lapse rate, snow depletion curve, snowmelt factors to provide a great scope in computing snowmelt and snowpack.

3- " . . . attempt to improve uncertainty propagation of these huge uncertainties in the input data."

We enhance the representation of the model performance in the calibration and validation stages. Table 4 was improved and Figure 2 developed to show these key aspects of modeling hydrologic processes.

We coincide with Reviewer 1 on the estimation of uncertainty and its propagation. The sensitivity analysis indeed leads to the need to address uncertainty propagation. However, we consider that this is a single document by itself. Ideas about the use of Machine Learning techniques have emerged in the team to address this particular and key aspect of water resources research in the Himalayas.

We appreciate reviewer-1 comment regarding the contribution of glacier melt. We provide more details in the methodology (section 2.4.3) as well as in the discussion. However, SWAT model does not differentiate melt water from glacier and non-glacier areas.

4- "... Beyond these coarse assumptions, the study is also somewhat limited in scope in such that it only considers changes in snowmelt and largely ignores melt water contributions from glacier melt..."

We valued Reviewr-1 comments and thanks to them we consider the present manuscript has been improved. Also, we consider this document contributes to better understand the hydrological implications of projected near-term and long-term snowmelt scenarios, represented by integrated snow cover and glaciers changes in response to a changing climate.

Below, you can see our general response to the review posted and the specific responses to the specific comments.

Specific Comments

Line 19: Suggest adding the size of the watershed considered in this study. This info is particularly of interest to assess how representative this study is for the entire Himalayan region.

We added the drainage area of the wastershed in the Abstract as is described in the Study Area section. [The main outlet point of the selected study area is Kasol, which drainage area is about 51,055 km2]. Line 26: Delete "approach" The sentence has been modified. Deleted Line 26. Line 25 ff.: It is not clear from the abstract what the

SWAT model is used for? The abstract could be streamlined to clearly state what the objective of the study is and what the different datasets and tools (e.g. SWAT) are used for. Also it would be good to clarify what snowmelt module was used in SWAT.

Now explained in the abstract.

[The primary objective of this study is to assess the role of elevations in the computation of snowfall (snowpack) and snowmelt in sub-catchments using the hydrological model Soil Water Assessment Tool (SWAT). SWAT in-built temperature index model was functionalized to assess the snowpack and snowmelt variations at sub-catchment-scale for both historical and projected near-term (2011-2030) periods using observed and Global Circulation Model (GCM) data sets.]

Line 74: "timing snowmelt"? Timing of snowmelt? How is glacier melt considered in the SWAT model? How is the model calibrated to consider seasonal changes in glacier mass balance, which includes quite a bit of snow?

This section has been re-structured and all the relevant and necessary information have been added. Line 70-73, 258-264.

[SWAT model provides melt water at each sub-catchment, which integrate the contributions of melt water from both snow and glaciers (Neitsch et al., 2011). The melt rate from a snowpack-glaciers varies in response to snowpack conditions (Fontaine et al., 2002). In SWAT modeling, the snowmelt and glacier melt were set up together as a linear function of the differences between the average of the snowpack and glacier temperature (Tsnowpack), and the maximum air temperature (Tmax) on a given day and the base or threshold temperature for the snowmelt (Eq. 7). Around 25% catchment area corresponded to glaciers and thus standard global coefficients related to glacier melting (e.g. glacier temperature) were incorporated to account melt release from glaciers, similarly applied by Grusson et al. (2015).]

Line 76: Wording! "understanding modeling complexities, mainly snowmelt induced: :

:"? Do you refer to snowmelt induced modeling complexities? Please revise!

Changed, see below

[In this study, our main concern is to compute an accurate snowpack and snowmelt in a snowmelt induced Himalayan catchment. The topography of the Satluj river catchment corresponded to extreme elevation ranges and steep slopes. Thus, multiple (maximum 10) elevation bands were formulated at each sub-catchment scale. The TLR and PLR have been calculated at each sub-catchment scale. Accordingly, the temperature and precipitation corresponding to each sub-catchment was adjusted as per the elevation differences for the computation of snowpack and snowmelt. The role of elevation bands in the computation of fractional snowpack and snowmelt at each sub-catchment scale is not accounted earlier. ]

Line 84: Delete "stream" in front of flows.

We will do. Authors have referred as streamflows in the past but are willing to address the reviewer-1 view.

Line 85: What percentage of the entire Himalayan region does the Satluj river basin represent?

10.21%

[The Satluj catchment is mainly covered by snow and glaciers, which covers around 30% of the total catchment area. The Satluj river catchment corresponded around 10.21% of the total Indian Himalayan region.]

Line 89: What is the percentage of glacier cover in each of the 14 sub-catchments?

Corrected

[The Satluj catchment is mainly covered by snow and glaciers, which covers around 30% of the total catchment area]

Line 110: What does SRES B2 stand for? This is not the most common way emission scenarios are referred to. Also it would be valuable to state what this scenario's main assumptions are.

This section has been re-structured and all the key features are explained accordingly. Line 138-142.

[In this study, the CGCM3.1/T63 atmospheric and sea-ice model outputs–namely, SRES B2 model experiment (Qiao et al., 2013) were used. CMIP3-SRES B2's daily temperature and precipitation were provided at 128x64 Gaussian grid (approximately 2.5° latitudes x 2.5° longitudes) (Thornton et al., 2009). The SRES B2 scenario emphasis on local rather than global solutions to economic, social and environmental stability. The SRES B2 experiment referred by various researcher in climate change study.]

Line 120: Can you clearly state how many meteorological stations and snow monitoring stations were included in this analysis and where these stations are located? Right now the manuscript suggests that the SWAT model uses only IPCC temperature and precipitation data, is then modeling the snowfall and snow accumulation and snowmelt. How much effort was spent to validate the quality and validity of the IPCC products in this river basin? There have been numerous studies concluding that snow accumulation in mountainous regions is highly variable. Without this uncertainty analysis in the input data the SUFI2 sensitivity analysis is worthless.

We used IMD observed gridded meteorological datasets. The figure 1 has been restructured. After that the gridded point has been spatially adjusted at each sub-catchment scale using TLR and PLR. This is a contribution of the present study. Additionally, in three sites have measured/gauged (points) precipitation data. We consider that for a 54,000km2 area the IMD datasets could be sufficient for the study, which approximates the referred drainage area at Kalso. It is noteworthy that the climatology of the Satluj River at the upstream catchments does not show any significant variations (except elevation differences and thus TLR and PLR that could be sensitive in

these catchments). Therefore, we adjusted the gridded precipitation and temperature datasets at each sub-catchment utilizing TLR and PLR.

[A gridded (at 1Ì£×1Ì£ degree resolution) daily precipitation, minimum and maximum temperature, humidity, wind speed and solar radiation datasets (1991-2008) product, obtained from the Indian Meteorological Department (IMD) and Indian Institute of Tropical Meteorology (IITM), was used. A total of 8 grids, within the limits of the catchment were selected (Fig. 1a). These datasets have been generated using more than 1800 observed precipitation and temperature gauges distributed across India.

The bias correction was on the GCM datasets was applied using observed datasets (Singh and Goyal, 2016)].

Line 126: How were historical scenarios of snowpack, snowmelt and other water balance components generated?

The information has been updated.

[The whole catchment was divided into 16 sub-catchments and 358 combining soil, LULC and slope. The water balance components were analyzed at each HRU scale and then the aggregated contribution of all the components can be accounted at each sub-catchment scale (Arnold et al., 1998).]

[In this study, the Soil Conservation Services (SCS) curve number (CN) was used to compute the water balance components.]

[The Satluj River catchment presents a very complex topography. Therefore, to account for the effect of steep slopes in the overall water balance computation, we employed a slope-adjusted modified CN-method (Arnold et al., 1998).]

[In this study, the linear reservoir approach was applied for the computation of snowpack and snowmelt by the fractional computation of snowpack and snowmelt (Fontaine et al., 2002). In this study, the cumulative of snowpack and snowmelt amount for the given elevation bands have been accounted at each sub-catchment scale (Neitsch

et al. 2011). The five basic snowmelt parameters such as degree day melt factor maximum (4.5 mm/d/°C, degree-day melt factor minimum (2.5 mm/ °C/d), snowfall temperature (0.0 °C), snowmelt temperature (0.5 °C) and snow water content for 100% snow cover (100mm) were incorporated in SWAT. Limited information is available on snowmelt and glacier hydrology over western Himalayan regions, thus, the regional/global parameter values were adopted (Neitsch et al. 2011; Fontaine et al., 2002).]

Line 131: The abstract states (line 21) that the basin was divided into 14 subcatchment. Here you state 16. Which one is correct?

We used 16 sub-catchments and this has been corrected everywhere in the manuscript.

Line 134: A little bit more information on the HRU delineation is needed. How many HRUs were derived for each sub-catchment? What curve numbers were assumed for the different soil types (which are not clearly defined in Figure 1!)? How were the 10 elevation bands derived (e.g. natural break, equal interval, equal area)? Why not use local information on the lapse rate to define the elevation bands?

In Table 1 we have provided the model simulation details like HRUs and other water balance components. Figure 1 has been modified and soil and LULC classes are explained now in a better way.

The local topographical characteristics were incorporated to define the elevation bands. This is explained through several extra points.

[To account for the effect of elevation and temperature variations, each sub-catchment was discretized using elevation bands. Fontaine et al. (2005) developed the methodology for snowmelt module in SWAT and suggested that the elevation bands contributed to improve hydrologic simulations. Thus, each sub-catchment has been discretized into a maximum of 10 elevation bands to improve the simulation of snowpack and snowmelt.
[Figure]

The elevation bands were defined by the averaged elevation and the associated (individual) contribution to the catchment area. In order to incorporate the temperature and precipitation variations across the sub-catchments, the TLR and PLR were computed and adjusted with respect to altitude of the corresponding sub-catchment (Neitsch et al., 2011).]

Line 140: More information on glacier cover in each sub-catchment is needed.

In this study, we accounted for glacier and snowmelt jointly. Thus, the total area covered by snowpack includes glacier cover.

Line 181: The SUFI2 approach might be useful for a watershed where the modeler has high confidence in the input data. Given that this study is using FAO soil maps (1x1 degree), IPPC gridded temperature and precipitation data which aggregate rainfall and snowfall input over a large region that does not take any spatial difference due to topography into account, the approach use of the SUFI2 is useless in the uncertainty in the input data is not properly addressed. This study would have much more scientific value if the authors would attempt to improve uncertainty propagation of these huge uncertainties in the input data.

The uncertainty in the modeling outcomes are incorporated through the detailed parameterization and sensitivity analysis. To account the uncertainty, around 20 calibration parameters were incorporated from each layer such as surface water, sub-surface water and groundwater. We ran numbers of simulations to get the optimized parameter values. However, we did not estimate the propagation of uncertainty, neither characterize the uncertainty associated with the forcings. We acknowledge that a more thorough study is needed to characterize sources of uncertainty and the propagation of it.

The calibration methodology has been re-explained and all the points, as asked, are added.

The SUFI2 algorithm assumes a large parameter uncertainty (or physically meaningful

range) occurring in response to data inputs to ensure the observed data fall into the 95% prediction uncertainty (95PPU) band during the first iteration (Abbaspour et al. 2011). During this iterative procedure, uncertainty progressive decrease is monitored though the changes of the p-factor and r-factor (Abbaspour et al., 2007). While the p-factor determines the percentage of simulated data falling into the observed-data range, the r-factor contributes to determine the uncertainty of the simulated variables and state variables when compared with observed data sets.

In SUFI2 optimization method, the old coefficient parameters (e.g. default parameter values) iteratively change and updated into a new array of coefficients during calibration to ultimately achieve the final set of parameters. It this way it improves the parameter coefficient values. In this study, around 600 simulations were conducted to account for the uncertain effect of the physics of the model and optimize the best-fitted coefficients as part of the calibration process.

In the sensitivity analysis, the statistical significance tests such as p-value test and t-stat were used to rank parameters from high- to non-sensitive.]

Line 216: It is not clear what stations were used to calculate the temperature difference and lapse rate for the elevation bands. The methods section does not contain any information on how many meteorological stations were included in the analysis.

See an explanation above.

[Total 8 grids which are falling inside/near to the catchment have been selected (Fig. 1a). These datasets have been generated using more than 1800 observed precipitation and temperature gauges distributed across India.

For the historical model simulation (1991-2008), the observed temperature and precipitation gridded datasets were spatially adjusted for each sub-catchment scale. The whole basin has been categorized into 16 sub-catchments based on the area-threshold method (Ficklin and Barnhart, 2014; Neitsch et al., 2011). Thus each grid point has

been spatially adjusted at the centroid of each sub-catchment. For the spatial adjustment of each grid point, at the centroid location of the sub-catchments, the TLR and PLR were computed (Singh and Goyal, 2016). The methodology of TLR and PLR will be explained in the next section.]

Line 252: The authors state clearly that SWAT is not capable to simulate glacier melt. However, 5 lines further down they claim that glacier melt was integrated with the snowmelt. Glacier melt presents a streamflow contribution from a storage reservoir, which cannot or should not be captured by the snowmelt routine. Once the accumulated precipitation is exhausted no more runoff can be contributed by snowmelt, however, this is typically the time when glacier melt contributions dominate. The authors state that there is a large number of glaciers in the basin (this should be clearly stated in the methods and basin description!) which can contribute glacier melt. Equation 7 is not capable to adequately capture glacier melt once the snow cover (snocov) is zero.

The reading is correct and has been clarified. So it explains that independent gracier melting is not parameterized but the joint snow and glacier melt. Based on the scope of the paper this satisfies our idea of reducing the complexity posed by independent snow and glacier melt dynamics. The information is explained.

[SWAT model provides melt water at each sub-catchment scale which includes the contribution of melt water from both snow and glaciers (Neitsch et al., 2011). The melt rate from a snowpack/glaciers varies in response to snowpack conditions (Fontaine et al., 2002). In SWAT modeling, the snowmelt and glacier melt were set up together as a linear function of the differences between the average of the snowpack and glacier temperature (Tsnowpack), and the maximum air temperature (Tmax) on a given day and the base or threshold temperature for the snowmelt (Eq. 7). Around 25% catchment area corresponded to glaciers and thus standard global coefficients related to glacier melting (e.g. glacier temperature) were incorporated to account for possible glaciers' melt contributions (Grusson et al. 2015).]

Line 302: "The Saltuj River drainage area is dominated by glacial hydrology, perma-nent ice sheets and seasonal well-packed snow." This sentence captures nicely the variability in snow and ice cover in the basin and illustrates the spatial and temporal variability of snowmelt processes that prevail in the basin, which cannot be captured by a melt rate model. Well-packed snow has a much greater density that freshly fallen snow and melts differently that new snow, aged snow or glacier ice. Since the authors do not provide any information on the HRU discretization of each sub-watershed I as-sume these spatial differences in snow and melt patterns are not accurately captured by the model.

In SWAT modeling, the glacier melt can be accounted using snowmelt parameters like depletion curves and glacier pack temperature and temperature lag factor. All these parameters are incorporated into the model. The parameter values are measured and also obtained through the literature survey.

[In this study, the linear reservoir approach was used to compute snowpack and snowmelt (Fontaine et al., 2002). The cumulative of snowpack and snowmelt amount for a given elevation bands were accounted at each sub-catchment (Neitsch et al. 2011).]

[To account for the effect of elevation and temperature variations, each sub-catchment was discretized into elevation bands. Fontaine et al. (2005) developed the method-ology for the snowmelt module in SWAT and suggested that the elevation bands im-proved the simulation of hydrologic processes. Thus, each sub-catchment has been discretized into a maximum of 10 elevation bands to improve the simulation of snow-pack and snowmelt. The elevation bands were defined by the averaged elevation and the associated (individual) contribution to the catchment area. In order to incorporate the temperature and precipitation variations across the sub-catchments, the TLR and PLR were computed and adjusted with respect to altitude of the corresponding sub-catchment (Neitsch et al., 2011).]

Line 394: The finding that low- to mid-latitude regions experience the most change from snowmelt to rainfall runoff is not surprising has been found in many other mountainous regions across the world (Foster et al. 2016, ERL; Klos et al. 2014 GRL).

We include views of Foster et al. 2016, ERL; Klos et al. 2014 GRL

Line 397: How would a change in elevation bands influence the results shown in Figure 7? Explained.

[Figure 7 illustrates the magnitude of change (shown as "% of change") in snowpack amount as a function of the fraction of elevation bands. At each sub-catchment, the snowpack differences have been computed and analyzed by comparing historical and near term scenarios. The results showed significant decrease in snowpack amount across all the snowpack-dominated sub-catchments. A 5% (minimum) to 42% (maximum) decrease in snowpack has been accounted across all the sub-catchments. This contributes to evidence increasing temperatures in snow cover and glacier melting over Himalayan catchments. The sub-catchments SB1, SB2, SB3, SB8 and SB16 correspond with the utmost decrease in snowpack amount (20% to 42%); whereas, the sub-catchments SB5, SB7, SB14 and SB15 SB16 showed a small to moderate decrease in snowpack amount (4% to 20%). Also, SBs, situated at high elevation areas showed larger changes or decreases in snowpack with respect to the moderate-elevation SBs. This illustrates a conspicuous warming trend at high elevation portions with respect to those at moderate and low elevation ranges.]

Figure 1: What do the black circles with the cross mean? Please ensure that each map shows a complete legend! The legend of the soil map is very uninformative. Could the different soils be grouped into know soil types or soil families? The 16 sub-catchments and their boundaries are hard to see. Please use a different line and text color for the catchment boundaries. Please add locations of glaciers within the basin. Also the locations of the three gauges mentioned in the methods cannot be found in any of the figures.

[Figure]

Figure 1 has been restructured.

Figure 2: Please correct the spelling of "snowmelt" in each graph. Please define what you mean by snowpack and snowmelt. Is snowpack the depth of the snowpack and snowmelt the snow water equivalent? Corrected. Snowpack (depth). Figure 4: The figure caption for this figure does not make sense. Graph (b) does not seem to show the cumulative variability in snowpack amount as stated in the caption.

Changed.

Figure 6: This figure is somewhat awkward. The labels for each plot are too big and cover in some instances the map. I would suggest putting each map into a grid/frame that is then clearly labeled with both a-j and the thematic title of the map.

Grid frame will reduce the spatial variations because at smaller scale the resolution of the figures will be distorted. The grid is not needed as their extent coordinate values we have already explained in the text (study area section). The captions are needed to explain each figure.

Fig. 7: Why are the bar graphs not sorted in increasing order from SB1-SB16 along the x-axis? SB5 is the last bar with the least change in snowpack but it is not clear what elevation band this sub-catchment belongs to. We considered that the current order reflects the geographical context rather than the numerical values assigned to each sub-catchment.

Anonymous Referee #2

General comments

The study aims to improve simulation of snowmelt runoff generation in Himalayas and to estimate its future changes.

The topic is interesting and relevant from both – scientific and water resources management perspective. The manuscript in its current form, however, does not clearly

formulate the novel scientific contribution. There are numerous studies presenting variability and change in snow accumulation and melt in Himalayas, including future projections, however it is not clear in which aspects is this study new. The calibration of a model and applying it then for simulations of future projections is by no means novel. The discussion of the results is missing, but this would be an opportunity to clearly show the contribution of the study with respect to existing results. More importantly, the justification of the model selection (used in the study) is not clear. I wonder how the SWAT model (CN approach) represents the dominant runoff processes in Himalayas. How does it account for glacier melt, snow accumulation and melt on steep slopes, the snow drift, energy balance, etc (i.e. processes and their changes related to snow melt generation)? Is the surface runoff really contributing more than 40% of annual runoff (table 1) in such large basin? I really missed to see some time-series (observed versus simulated) and process-based interpretation (with justification) of results. I'm sorry, but I do not recommend to publish the study in its current form. At least a significant revision is needed. We appreciate Reviewer-2's valuable comments toward improving the present document. We made changes to the narrative, in terms of the justification, the clarification of why this study is relevant and contribute to the current understanding of Himalayan diagnostic and prognostic hydrology (referenced in multiple studies), and the calibration/validation implemented in the present study. Furthermore, we updated the Discussion section, including a thorough analysis of the information emerged from the modeling data and supported by current and past literature. Below you will see the statements and/or paragraphs written, as examples of how we address the specific comments of the reviewers within the new version of the text. We address reviewer-2 comments in four sections based on the review as follows:

1-"clearly formulate the novel scientific contribution"

The justification regarding the novelty of the present research work is explained in the manuscript as follows.

[In this study, our main concern is to compute an accurate snowpack and snowmelt in a

snowmelt induced Himalayan catchment. The topography of the Satluj river catchment corresponded to extreme elevation ranges and steep slopes. Thus, multiple (maximum 10) elevation bands were formulated at each sub-catchment scale. The TLR and PLR have been calculated at each sub-catchment scale. Accordingly, the temperature and precipitation corresponding to each sub-catchment was adjusted as per the elevation differences for the computation of snowpack and snowmelt. The role of elevation bands in the computation of fractional snowpack and snowmelt at each sub-catchment scale is not accounted earlier.] 2-"the justification of the model selection (used in the study) is not clear"

The growing use worldwide of SWAT, together with available resources, modular design and flexibility are just some architectural features that favor the use of such land surface hydrology model in the region. From a conceptual modeling perspective, SWAT is a platform that allowed our team to sun a sensitivity test for TLR and PLR at sub-catchment scale. This is particularly keen when reviews such as the one lead by Dimri et al (2015) suggest that "understanding the different mechanisms that contribute to the temporal evolution of the families of West Disturbance storms at regional scale and to understand the complex spatial and temporal variability of precipitation and associated weather at scales relevant for hazard mitigation and decision making". The local scale also referred in this document, highlights the value of understanding hydrological processes and drive prognostic tools to better understand at sub-catchment scales. We also improved the justification to the present work in the manuscript.

[SWAT is a physical parameter based deterministic hydrological model, which is able to compute streamflow and snowmelt at sub-catchment scale in the daily time steps (Fontaine et al., 2002). SWAT has already proven their capability in the computation snowpack and snowmelt (Ficklin and Barnhart, 2014; Neupane et al., 2014; Fontaine et al., 2002).] 3- "How does it account for glacier melt, snow accumulation and melt on steep slopes, the snow drift, energy balance, etc. (i.e. processes and their changes related to snow melt generation)?"

[Figure]

We address this question by improving the description of the model and its implementation in the area of study. Below you will see a glance of the narrative developed in the updated version of the methodology.

[In this study, the main hydrological processes included interception, infiltration, runoff, evapotranspiration, lateral flow and percolation. The details about the physical and hydrological principles and parameters of SWAT are fully described in the SWAT user manual (Neitsch et al. 2011). In this study, the linear reservoir approach was applied for the computation of snowpack and snowmelt by the fractional computation of snowpack and snowmelt (Fontaine et al., 2002). Further, the accumulation of snowpack and snowmelt amount for the given elevation bands have been accounted at each sub-catchment scale (Neitsch et al. 2011). The five basic snowmelt parameters such as degree day melt factor maximum (4.5 mm/d/°C, degree-day melt factor minimum (2.5 mm/ °C/d), snowfall temperature (0.0 °C), snowmelt temperature (0.5 °C) and snow water content for 100% snow cover (100mm) were incorporated in SWAT. A limited number of modeling studies have explored snowmelt and glacier hydrology over western Himalayan regions. Thus, the regional/global parameter values adopted here represent values available for the model simulation (Neitsch et al. 2011; Fontaine et al., 2002). In this study, these snowmelt parameters are considered as model calibration parameters. Based on the modified CNs, the stream flow is generated at each sub-catchment scale and then the aggregation was done at the outlets. Similarly the other water balance components were simulated. Based on the GCMs, the daily temperature and precipitation were incorporated into SWAT and then near tern scenarios were computed. In this study, SWAT based Temperature Index Model (TIM) has been used to compute the snowpack and snowmelt at the sub-catchment scale utilizing observed and GCM based daily minimum and maximum temperature (Fontaine et al., 2002). The seasonal degree-day melt factor (e.g. maximum and minimum) global coefficient values obtained from the literature survey were employed in the model for the computation of snowmelt (Neitsch et al., 2011; Fontaine et al., 2002; Anderson, 1976). In Himalayan region, the degree-day melt factor varies from 1.5 mm/°C/d to 4.7 mm/°C/d

(Fontaine et al., 2002). To account the effect of elevation and temperature variations, each sub-catchment was divided elevation bands. Fontaine et al. (2005) developed the methodology for snowmelt module in SWAT and suggested that the elevation bands improved the simulation. Thus, each sub-catchment has been categorized into maximum 10 number of elevation bands to improve the simulation of snowpack and snowmelt. The elevation bands are defined based on their mean elevation and the proportion of the sub-catchment area they encompass.]

Please refer to sections 2.4.1 to 2.4.3

4- "Is the surface runoff really contributing more than 40% of annual runoff (table 1) in such large basin?"

The Satluj River basin's precipitation is around 1200mm per year, which contributes to perennial streamflows in the basin. Surface runoff is the main source of streamflow in Satluj river. We don't think that this is very high in case of Satluj River.

Please also note the supplement to this comment:
http://www.hydrol-earth-syst-sci-discuss.net/hess-2016-689/hess-2016-689-AC1-supplement.pdf

**Fig. 1**: (a) Study area map of Satluj river catchment (up to Kasol gauge).

[Figure]

(b) Landuse/landcover map (LULC) and soil map of the study area.

[Figure]

**Fig. 1.** Figure 1
Interactive
comment

[Figure]

**Fig. 2 (a)**: Scatter plots of daily calibration (1991-2000) and validation (2001-2008) at three outlets: (a) Rampur calibration, (b) Rampur validation. (c) Suni calibration, (d) Suni validation, (e) Kasol calibration, (f) Kasol validation.

**Fig. 2.** Figure 2a
Interactive
comment

**Fig. 2 (b)**: Sub-catchment and annual variability in snowpack and snowmelt (annual average) for the year 1991 to 2008.

[Figure]

**Fig. 3.** Figure 2b
Interactive
comment

**Fig. 3**: Distribution of average temperature over the sub-watershed's centroid elevation (in chronological order); (a) winter season and (b) summer season.

(a)

[Figure]

(b)

**Fig. 4.** Figure 3
Interactive
comment

**Figure 4**: (a-h) Sub-catchment wise variations in fractional snow covers computed as per elevation bands in during whole time series (1991-2030).

[Figure]

**Fig. 5.** Figure 4

**Figure 5 (b)**: Variability in snowpack amount during different time series sets computed at each sub-catchment scale.

[Figure]

**Fig. 6.** Figure 5
* * *
Interactive
comment

**Fig. 6**: Changes and inter annual comparisons in average annual (a) precipitation, (b) snowpack/snowfall, (c) snowmelt, (d) water yield (due to snow) and (e) total water yield (snowmelt and rainfall runoff) over the study area in different temporal climate domains (1991-2030).

[Figure]

**Fig. 7.** Figure 6

[Figure]

**Fig. 7**: Percentage of change in snowpack amount (average annual) over different sub-catchments of Satluj river.

[Figure]

**Fig. 8.** Figure 7

**Table 4**: Model calibration and validation results as per SUFI method on the daily and monthly basis analysis.

| Outlet Station | Calibration (1991 - 2000) | | | | | | | |
|---|---|---|---|---|---|---|---|---|
| | Daily | | | | Monthly | | | |
| | p-factor | r-factor | $R^2$ | NSE | p-factor | r-factor | $R^2$ | NSE |
| Rampur | 0.46 | 1.89 | 0.75 | 0.61 | 0.41 | 1.90 | 0.71 | 0.64 |
| Kasol | 0.57 | 1.50 | 0.76 | 0.63 | 0.57 | 1.57 | 0.78 | 0.67 |
| Suni | 0.52 | 1.60 | 0.72 | 0.59 | 0.49 | 1.43 | 0.73 | 0.60 |
| Outlet Station | Validation (2001 - 2008) | | | | | | | |
| | Daily | | | | Monthly | | | |
| | p-factor | r-factor | $R^2$ | NSE | p-factor | r-factor | $R^2$ | NSE |
| Rampur | 0.43 | 1.89 | 0.62 | 0.54 | 0.45 | 1.92 | 0.65 | 0.55 |
| Kasol | 0.52 | 1.67 | 0.71 | 0.59 | 0.60 | 1.62 | 0.73 | 0.61 |
| Suni | 0.52 | 1.72 | 0.65 | 0.58 | 0.58 | 1.52 | 0.71 | 0.60 |

**Fig. 9.** Table 4